# Calcium Signaling Pathway Is Involved in the Shedding of ACE2 Catalytic Ectodomain: New Insights for Clinical and Therapeutic Applications of ACE2 for COVID-19

**DOI:** 10.3390/biom12010076

**Published:** 2022-01-05

**Authors:** Artemio García-Escobar, Silvio Vera-Vera, Alfonso Jurado-Román, Santiago Jiménez-Valero, Guillermo Galeote, Raúl Moreno

**Affiliations:** 1Cardiology Department, Interventional Cardiology Section, University Hospital La Paz, 28046 Madrid, Spain; geneser_hall@hotmail.com (S.V.-V.); alfonsojuradoroman@gmail.com (A.J.-R.); sjvcardio@yahoo.es (S.J.-V.); ggaleote1@gmail.com (G.G.); raulmorenog@hotmail.com (R.M.); 2Instituto de Investigación Hospital La Paz (IDIPAZ), 28029 Madrid, Spain; 3Centro de Investigación Biomédica en Red Enfermedades Cardiovasculares (CIBERCV), Instituto de Salud Carlos III, 28029 Madrid, Spain

**Keywords:** angiotensin-converting enzyme 2, calmodulin, COVID-19, soluble ACE2, ACE2 shedding, soluble catalytic ectodomain of ACE2, calcium signaling, vitamin D, COVID-19 thromboembolic events, neuropilin-1

## Abstract

The angiotensin-converting enzyme 2 (ACE2) is a type I integral membrane that exists in two forms: the first is a transmembrane protein; the second is a soluble catalytic ectodomain of ACE2. The catalytic ectodomain of ACE2 undergoes shedding by a disintegrin and metalloproteinase domain-containing protein 17 (ADAM17), in which calmodulin mediates the calcium signaling pathway that is involved in ACE2 release, resulting in a soluble catalytic ectodomain of ACE2 that can be measured as soluble ACE2 plasma activity. The shedding of the ACE2 catalytic ectodomain plays a role in cardiac remodeling and endothelial dysfunction and is a predictor of all-cause mortality, including cardiovascular mortality. Moreover, considerable evidence supports that the ACE2 catalytic ectodomain is an essential entry receptor for severe acute respiratory syndrome coronavirus 2 (SARS-CoV-2) infection. Additionally, endotoxins and the pro-inflammatory cytokines interleukin (IL)-1β and tumor necrosis factor-alpha (TNFα) all enhanced soluble catalytic ectodomain ACE2 shedding from the airway epithelia, suggesting that the shedding of ACE2 may represent a mechanism by which viral entry and infection may be controlled such as some types of betacoronavirus. In this regard, ACE2 plays an important role in inflammation and thrombotic response, and its down-regulation may aggravate COVID-19 via the renin-angiotensin system, including by promoting pathological changes in lung injury. Soluble forms of ACE2 have recently been shown to inhibit SARS-CoV-2 infection. Furthermore, given that vitamin D enhanced the shedding of ACE2, some studies reported that vitamin D treatment is associated with prognosis improvement in COVID-19. This is an updated review on the evidence, clinical, and therapeutic applications of ACE2 for COVID-19.

## 1. Introduction

On December 2019, at Wuhan, China had reported the first cases of pneumonia due to a novel coronavirus that was identified as a betacoronavirus in the same subgenus as the severe acute respiratory syndrome-related coronaviruses (SARSr-CoVs) attachment [1].The Coronaviridae Study Group of the International Committee on Taxonomy of Viruses classified this this virus as severe acute respiratory syndrome coronavirus 2 (SARS-CoV-2) [2]. SARS-CoV-2 is the seventh coronavirus that is known to cause human disease [3]. The World Health Organization declared COVID-19 to be a pandemic in March 2020 [4]. The SARS-COV-2 share a 79.6% sequence identity to severe acute respiratory syndrome coronavirus (SARS-CoV-1) and is 96% identical to a bat coronavirus [5]. SARS-CoV-1 attaches to the target cell through the interaction of the spike glycoprotein with its receptor, the angiotensin-converting enzyme 2 (ACE2) [6]. Moreover, many studies demonstrated that the catalytic ectodomain of ACE2 (subdomain I) is an essential entry receptor for SARS-CoV-2 infection [5,7]. 

ACE2 binding triggers a spike protein-mediated fusion of the viral envelope with the cell plasma membrane or intracellular endosomal membranes. The spike protein is synthesized as a single polypeptide that is cleaved by the cellular protease furin into S1 and S2 subunits in the endoplasmic reticulum and then further processed by transmembrane serine protease 2 (TMPRSS2) on target cells [8,9]. The S1 subunit contains the receptor binding domain (RBD), which binds to ACE2, while S2 mediates virus–cell fusion [9]. Although younger patients are thought to have a good prognosis, older patients and those with chronic underlying conditions, such as diabetes (DM) or hypertension, may have worse outcomes which may progress to acute respiratory distress syndrome (ARDS) or end-organ failure [10]. Furthermore, cohort studies have reported higher mortality in severe COVID-19 than in non-COVID-19 severe pneumonia [11]. The laboratory findings of a higher neutrophil count, D-dimer, ferritin, c-reactive protein (CRP), and low lymphocyte count have been reported in severe COVID-19 and were associated with unfavorable prognosis [10,12,13]. Hence, understanding the physiology of ACE2 could provide a better understanding of the cardiovascular disease and SARS-CoV-2 physiopathology.

## 2. ACE2 Molecular Structure

The first known human homolog of the angiotensin-converting enzyme (ACE) was discovered in 2000 by two independent groups and was designated as ACE2 [14,15]. The ACE2 gene contains 18 exons, interspersed with 17 introns, localized to chromosome X, position p22.2. The ACE2 is a type I integral membrane protein formed of 805 amino acids that contains two domains [14,15,16]. The first is a transmembrane domain (amino acids 741–763) [15] and the second is an extracellular region, also known as the ectodomain (amino acids 1–740) [16]. The extracellular region is composed of two subdomains: [16] the first is a zinc metallopeptidase domain (amino acids 19–611) that is the single catalytic domain of the ACE2, [14,15] which is 42% identical to each of the two catalytic domains in ACE; the second is located at the C terminus (amino acids 612–740) and is 48% identical to a transporter protein known as collectrin [16]. ACE2 activity is unaffected by 10 μM lisinopril, enalaprilat, or captopril, but activity was completely inhibited by 10 μM of ethylenediaminetetraacetic acid (EDTA) [14]. This supports that the ACE2 is a metalloprotease of the M2 family with a consensus sequence HEXXH at amino acid positions 374–378, but with a substrate and inhibitor specificity distinct from ACE [14]. The ACE2 mRNA is localized to the endothelium of vessels and to a minor extent in vascular smooth muscle cells [15]. Moreover, ACE2 mRNA expression is greater in the heart, kidney, and testis, and with moderate levels in the colon, small intestine, and ovary [14]. Similarly, ACE2 expression is increased in alveolar type II cells, [17] and there is some ACE2 expression at the apical surface of the airway epithelia [18].

## 3. The ACE2 Catalytic Ectodomain Undergoes Shedding

The catalytic active ectodomain of ACE2 is located at the extracellular region and undergoes shedding by a disintegrin and metalloproteinase domain-containing protein 17 (ADAM17), also known as a tumor necrosis factor alpha-converting enzyme, which was first discovered as the sheddase responsible for the cleavage of the inflammatory cytokine tumor necrosis factor-alpha (TNF-α) [19]. Thereafter, studies demonstrated that it also exerts functions as a protease that is up-regulated in heart failure (HF) [20]. This process releases a soluble form of ACE2 (Figure 1). The ACE2 cleavage region is localized near the predicted transmembrane domain, between amino acids 716 and 741 [18]. Therefore, ACE2 can be found in two forms: the first is as a protein anchored to the plasma membrane through a transmembrane domain near the C terminus that can be measured as ACE2 mRNA expression; [14] the second is as a soluble catalytic ectodomain of ACE2, which lacks the cytosolic and transmembrane domains, also known as the soluble ACE2 that can be found in the plasma and other body fluids. This can be measured as soluble ACE2 plasma activity [14].

## 4. ACE2 and Its Unique Cleavage of Key Vasoactive Peptides: The Essential Role for the Renin-Angiotensin System

ACE2 activity has a strong pH dependence under acidic conditions, such that the enzyme is almost inactive at pH 5.0 and has optimal activity at pH 6.5 [21]. However, ACE2 maintains substantial catalytic activity under basic conditions (pH 7–9). ACE2 proteolytic activity is greatly enhanced by high concentrations of chloride or fluoride, but it is not enhanced in the presence of bromide ion [21]. The carboxy-terminal leucine of the angiotensin I (AT1) is hydrolyzed by the ACE2 to generate angiotensin 1-9 (AT1–9), which is converted to smaller angiotensin peptides by ACE. ACE2 can also cleave des-Arg bradykinin, neurotensin 1–13, kinetensin, angiotensin II (AT2), AT1, apelin-13, beta-casomorphin, dynorphin A 1–13, and ghrelin, but not bradykinin or 15 of the other vasoactive and hormonal peptides that have been tested [15,21]. Moreover, des-Arg bradykinin is involved locally in vessel dilation through its binding to the B1 receptor that is expressed under conditions of tissue damage or inflammation. This is consistent with the role of ACE2 in the local regulation of vasomotor tone through both AT1 and des-Arg bradykinin cleavage [15]. In contrast, ACE cleaves the vasodilator bradykinin, which acts systemically through the B2 receptor. Whereas kinetensin stimulates mast cell degranulation and vascular permeability, the degradation of these peptides by ACE2 may serve to modulate these activities [15]. Moreover, ACE2 is expressed in the luminal surface of differentiated small intestinal epithelial cells and is a modulator of gut microbial ecology, innate immunity and transmissible susceptibility to colitis [22]. In addition, ACE2 regulates the dietary amino acid homeostasis and is essential for the intestinal uptake of tryptophan [23]. 

Prospective cohorts have demonstrated that a high equilibrium and circulating levels of angiotensin 1–7 (AT1-7)/AT2 ratios were associated with improved survival, lower hospitalization duration, and better New York Heart Association (NYHA) class irrespective of left ventricular ejection fraction (LVEF) [24,25]. This notwithstanding, individual AT1-7 and AT2 peptide levels failed to predict all-cause mortality or hospitalization duration due to HF [25]. Doses of 100–1200 μg/kg of a recombinant version of the catalytic ectodomain of human ACE2, also known as the human recombinant soluble ACE2 (hrsACE2), are well tolerated by healthy human subjects [26]. This hrsACE2 was given as an intravenous infusion over 30 minutes and had a terminal half-life of about 10 hours [26]. The administration of hrsACE2 effectively normalized elevated AT2 while increasing AT1-7 and AT1–9 in patients with HF [24]. Significantly, when the effect of hrsACE2 in the subgroup of chronic HF patients treated with angiotensin receptor blockers was compared with chronic HF patients treated with angiotensin converting enzyme inhibitors (ACEIs), the AT2 was found to be lowered, thereby increasing the AT1-7/AT2 ratio, but the AT1-7 was only mildly increased, with a relatively dominant increase in plasma angiotensin 1–5 (AT1-5). Therefore, ACEIs reduced AT1-7 degradation, whereas hrsACE2 increased AT1-7 formation [24].

The chymase is a chymotrypsin-like serine protease that is synthesized and stored in the cardiac mast cells and is an alternative source of AT2 that is increased in failing human hearts [27]. A study tested the effects of hrsACE2 on the chymase of explanted human hearts with dilated cardiomyopathy, demonstrating that the AT2 was effectively converted into AT1-7, with an increasing in the AT1-7/AT2 ratio [24]. AT2 levels were concomitant with increased chymase levels and activity, despite the use of ACEIs, but were completely suppressed by hrsACE2 [24]. AT1-7 has a very short half-life (<9 seconds), and the release of a soluble ectodomain of ACE2 from the vascular endothelium may serve to alter systemic AT1-7 concentrations and the relative peripheral balance of ACE2/ACE [28]. Hence, the ACE/AT2/angiotensin I receptor (AT1R) axis exerts a detrimental effect on the cardiovascular system whereas the ACE2/AT1-7/Mas axis exerts cardiovascular protection [29]. 

## 5. The Soluble ACE2 Plasma Activity and Its Association with Cardiac Remodeling, Endothelial Dysfunction and Prognosis in Heart Failure and Cardiovascular Disease

In the last years, studies showed that increasing soluble ACE2 plasma activity is associated with clinical HF in patients with ischemic and without ischemic cardiomyopathies, and is correlated with increasing B-type natriuretic peptide levels and a worsening NYHA functional class [28,30]. Moreover, higher soluble ACE2 activity correlates with elevated plasma asymmetric dimethylarginine, a marker of oxidative stress-mediated endothelial dysfunction in patients with HF [31]. The Finnish Diabetic Nephropathy Study showed that soluble ACE2 increases in patients with type 1 DM and vascular complications, such as albuminuria and coronary artery disease (CAD) [32]. A study reported that elevated soluble ACE2 in acute myocardial infarction (MI) is correlated with infarct size and is associated with the occurrence of left ventricular remodeling [33]. Soluble ACE2 is increased in atrial fibrillation (AF) and is associated with abnormal left atrial (LA) electroanatomic mapping parameters, which reflects an advanced LA structural remodeling [34]. Moreover, in patients with aortic stenosis, the elevated soluble ACE2 plasma activity was an independent predictor of all-cause mortality. In addition, increased soluble ACE2 plasma activity was associated with reduced myocardial ACE2 gene expression and severe myocardial fibrosis [35]. Interestingly, the Prospective Urban Rural Epidemiology (PURE) study, a prospective study (*n* = 10,743), showed that being male, having a higher body mass index, DM, older age, higher low-density lipoprotein (LDL) cholesterol, smoking, and higher blood pressure were all associated with increased levels of circulating plasma ACE2. Additionally, a higher plasma ACE2 concentrations were associated with a greater risk of incident HF, MI, stroke, and DM, as well with increased risk of total deaths with similar increases in cardiovascular and non-cardiovascular deaths [36]. Although many studies demonstrated that the shedding of the ACE2 catalytic ectodomain plays a role in the process of cardiac remodeling and endothelial dysfunction, it is unclear whether the presence of circulating soluble ACE2 in plasma indicates the cause or effect of an adaptive or maladaptive physiological process operating in HF and cardiovascular diseases. Notwithstanding, at the present time, it is proven by many studies that high soluble ACE2 plasma activity is a predictor of major adverse cardiovascular events (MACE), cardiovascular mortality, and all-cause mortality [28,30,32,33,34,35,36].

## 6. COVID-19 and Its Association with Thromboembolic Events

It is well known by many cohort studies that the incidence of venous thromboembolism (VTE) is higher among patients with severe COVID-19, relative to its presence among patients with respiratory failure related to other causes [37,38]. A meta-analysis (*n* = 8271) revealed that the overall deep vein thrombosis (DVT) rate was 20%, in the intensive care unit (ICU), it was 28%, and postmortem, it was 35%. The overall pulmonary embolism (PE) rate was 13%, in the ICU, it was 19%, and postmortem, it was 22%. The overall arterial thromboembolism rate was 2%, and in the ICU, it was 5%. Moreover, the pooled odds of mortality were 74% higher among patients who developed thromboembolism compared to those who did not, with an odds ratio (OR) of 1.74; 95% confidence interval (CI) 1.01–2.98 (*p* = 0.04) [37]. A Spanish study of multiple hospitals during the first COVID-19 outbreak (*n* = 74,814) showed that PE in COVID-19 patients at emergency department (ED) presentation was about 0.5%. The standardized incidence of PE in the COVID-19 population resulted in 310 per 100,000 person-years vs. 35 per 100,000 person-years in non-COVID-19 population (OR 8.95; 95% CI 8.51–9.41). Thus, even though the PE in COVID-19 patients at ED presentation was low, the incidence is approximately ninefold higher than in the general (non-COVID-19) population. Furthermore, several characteristics in COVID-19 patients were independently associated with PE; the strongest direct associations were D-dimer levels >1000 ng/mL and chest pain, and chronic HF had an inverse associatio [39]. In contrast, the International Society on Thrombosis and Haemostasis diagnostic criteria for overt disseminated intravascular coagulation (DIC) included a category of coagulopathy associated with sepsis, termed sepsis-induced coagulopathy [40]. The coagulopathy in COVID-19 differs from DIC in that fibrinogen levels are elevated (in classic DIC, they should be low and are often <1 g/L), thrombocytopenia is mild (platelet counts in DIC are usually <50 × 109/L), and the prothrombin time (PT) is only slightly prolonged (1 to 2 seconds vs. 3 to >6 seconds in DIC due to the consumption of procoagulant factors) [41]. Therefore, the SARS-CoV-2 infection has a unique laboratory signature, including elevated fibrinogen and D-dimer with mild thrombocytopenia. This prompted some investigators to prefer the term COVID-19-associated coagulopathy [41]. Given the proven association of thrombosis in COVID-19, many trials tested the use of anticoagulation (AC) treatments in patients with COVID-19, especially in those with severe COVID-19. A cohort study reported that no difference in the 28-day mortality was found between heparin users and nonusers in the COVID-19 group (30.3% vs. 29.7%, *p* =  0.910). This notwithstanding, when patients had D-dimer levels greater than 3.0 μg/mL, those receiving unfractionated heparin (UFH) had lower mortality rates than those not receiving unfractionated heparin (32.8% vs. 52.4%, *p* = 0.017), suggesting that only patients with markedly elevated D-dimer may benefit from anticoagulant treatment [11]. Furthermore, a retrospective analysis of hospitalized patients with COVID-19 in five New York city hospitals (*n* = 4389) demonstrated that therapeutic AC was associated with a 47% reduction in the hazard ratio (HR) of in-hospital mortality 0.53 (95% CI: 0.45 to 0.62; *p* < 0.001) compared with no AC. Similarly, prophylactic AC was associated with lower mortality compared with no AC. Moreover, therapeutic AC was associated with a 31% reduction in the hazard of intubation (HR: 0.69; 95% CI: 0.51 to 0.94; *p* = 0.02), while prophylactic AC was also associated with a reduced incidence of intubation compared with no AC. Additionally, the administration of AC within 48 hours of admission showed that therapeutic AC was associated with a 14% reduction in hazard of mortality compared with prophylactic AC, though this did not reach statistical significance (*p* = 0.08). This notwithstanding, the proportion of patients with bleeding events after the initiation of AC treatment was highest in patients on therapeutic AC (3.0%) compared with patients on prophylactic AC (1.7%) and no AC (1.9%). Bleeding rates were higher in those on low molecular weight heparin (LMWH) compared with direct oral anticoagulants (DOACs) (2.6% vs. 1.3%). Bleeding rates were higher in those on UFH compared with LMWH (1.7% vs. 0.7%). Therefore, the study confirmed that AC lowered mortality among hospitalized COVID-19 patients [42]. In contrast, the trials of anti-thrombotic therapy for adults hospitalized with COVID-19 (ACTIV-4), antithrombotic therapy to ameliorate the complications of COVID-19 (ATTACC), and the randomized, embedded, multifactorial adaptive platform trial for community-acquired pneumonia (REMAP-CAP) paused their enrolment temporally because of the futility of attempting to treat severe COVID-19. Thereafter, they restarted the enrolment into the moderate COVID-19 stratum, which included patients not receiving ICU level of care. In patients with moderate disease, regardless of D-dimer concentration, therapeutic AC decreased the number of days on organ support. This suggested that AC worked best when started early in the disease course before patients became critically ill [43]. Nonetheless, further studies are warranted to assess the optimal AC regimens and how to choose the appropriate subset of patients that will benefit from AC. 

## 7. ACE2 Plays an Important Role in Thrombogenic and Inflammatory Activity

Pre-clinical studies showed that the lower activity of ACE2, AT1-7, and ACE2/ACE ratio were associated with higher thrombus formation in vessels [44,45]. Moreover, treatment with a small-molecule xanthenone ACE2 activator increased the time for thrombus formation by 45%, whereas the platelet attachment to vessels was reduced and the thrombus area decreased by 60% [46]. Thus, the ACE2/AT1-7/Mas axis exerts an antithrombotic effect [44,45]. The ACE2-catalyzed hydrolysis of AT2 produces AT1-7 [21]. Conversely, AT2 increases the production and secretion of plasminogen activator-inhibitor type 1 from endothelial and smooth muscle cells and the augmentation of tissue factor expression, thereby enhancing the activity of the coagulation system [46]. Thus, AT2-mediated microvascular thrombosis involves the activation of the AT2 receptor [46]. In contrast, AT1-7 is a potent inhibitor of thrombus formation in Mas-positive cells [45]. As a result, AT1-7 activates endothelial nitric oxide synthase (NOS) through an Akt-dependent mechanism and attenuates Nicotinamide adenine dinucleotide phosphate (NADPH) oxidase via the Mas receptor, playing an essential role in maintaining vascular integrity and endothelial function [47]. Additionally, AT1-7 activates the Mas receptor and exerts various effects, the majority of which antagonize the effects of AT2. These effects include the activation of the phosphatidylinositol 3-kinase–Akt–endothelial nitric oxide synthase pathway, the inhibition of protein kinase C–p38 mitogen-activated protein kinase pathways, and the inhibition of collagen expression to limit cardiac fibrosis [29]. Therefore, the ACE2/AT1-7/Mas axis offers cardiovascular protection by providing vasodilator, antioxidant, anti-thrombotic, anti-fibrotic and anti-hypertrophic effects, whereas the ACE/AT2/AT1R axis exerts the opposite effect by providing vasoconstriction, endothelial injury, thrombosis, fibrosis, and hypertrophy effects [29]. On the other hand, interleukin (IL)-6 promotes AT1R expression by sensitizing the vascular wall to AT2-dependent signaling mechanisms. The activated cells can release cytokines, including IL-6, which promote the expression of adhesion molecules and induce endothelial activation, inflammatory cell infiltration, and vascular inflammation [48]. Pre-clinical studies with ARDS models have shown that the loss of ACE2 expression resulted in enhanced vascular permeability, increased lung edema, neutrophil accumulation, and worsened lung function [49]. Interestingly, a small study of patients with acute respiratory distress syndrome showed that AT2 was higher in non-survivors than in survivors, the use of hrsACE2 was well tolerated given by three days with twice-daily infusions of 0.4 mg/kg. The result was a decreased of AT2 level concentrations, whereas surfactant protein D, AT1-7, and AT1-5 concentrations were increased [50]. SARS-CoV-2 binds to the catalytic ectodomain of ACE2, and the subsequent membrane fusion and virus entry into the cell leads to the down-regulation of these receptors. This ACE2 down-regulation, induced by the cell entry of the virus, may be particularly detrimental in patients with preexisting ACE2 deficiency due, for example, to older age, DM, and hypertension [51]. Furthermore, these factors were associated with the development of ARDS in many studies of patients with COVID-19, such as the cohorts of Wuhan and Lombardy [10,52]. In this regard, a postmortem study that compared COVID-19 with ARDS secondary to influenza A H1N1 infection reported that angiogenesis was 2.7 times as high, and alveolar-capillary microthrombi were nine times as prevalent in patients with COVID-19. These findings suggest the presence of widespread thrombosis with microangiopathy and endothelial injury in COVID-19 [53]. On the other hand, a recent preclinical study demonstrated that elevated glucose levels enhance SARS-CoV-2 replication and cytokine expression in monocytes [54]. Interestingly, they showed that SARS-CoV-2-infected monocytes promote T cell dysfunction and lung epithelial cell death, [54] and this could explain why diabetic patients have a worse prognosis. Additionally, another recent study revealed that the SARS-CoV-2 spike protein alone can damage endothelium, with the damage manifested by impaired mitochondrial function and endothelial NOS activity, but that increased glycolysis, a dysregulated renin-angiotensin system, due to ACE2 reduction may exacerbate endothelial dysfunction, leading to endotheliitis [55]. Therefore, in the setting of enhanced ACE2 deficiency produced by the viral invasion, the marked dysregulation of the ACE2/AT1-7/Mas axis would contribute to enhancing the progression of inflammatory and thrombotic processes [51]. Thus, ACE2 plays an important role in inflammation, and its down-regulation may aggravate COVID-19 via the renin-angiotensin system, including by promoting pathological changes in lung injury and involving inflammatory responses in conjunction with the release of cytokines, especially IL-6 [56]. Nevertheless, a meta-analysis with direct systematic comparisons of COVID-19 with other critical illnesses associated with elevated cytokine concentrations (*n* = 1245) reported that in patients with severe or critical COVID-19, the pooled mean serum IL-6 concentration was 36.7 pg/mL. Mean IL-6 concentrations were nearly 100 times higher in patients with cytokine release syndrome (3110.5 pg/mL), 27 times higher in patients with sepsis (983.6 pg/mL), and 12 times higher in patients with ARDS unrelated to COVID-19 (460 pg/mL) [57]. This suggests that IL-6 does not play the key role in the development of inflammatory and thrombosis processes in COVID-19, but rather is just one more factor added to the COVID-19 physiopathology. Hence, the endothelial injury seen in COVID-19 is probably the key factor for understanding the physiopathology of SARS-CoV-2 infection.

## 8. The Calcium Signaling Pathway Is Involved in ACE2 Release: The Role of Vitamin D in COVID-19?

The calcium signaling pathway is involved in the catalytic ectodomain of the ACE2 shedding process regulated by calmodulin (CaM) [18,20]. CaM is a prototypical and versatile calcium sensor with EF-hands (helix-loop-helix structural domain) as its high-affinity calcium. CaM is a ubiquitous calcium-binding protein that is known to bind other transmembrane proteins and regulate their cell surface expression [58,59]. Moreover, CaM binds the cytoplasmic sequences of the domain of L-selectin [60]. Similarly, CaM was shown to bind to the membrane-proximal cytoplasmic sequences of the platelet membrane glycoprotein (GPVI) [61]. The juxtamembrane cytoplasmic sequence of ACE2 is homologous to the membrane-proximal sequences of GPVI and L-selectin [60,61]. Furthermore, a computational analysis of the cytoplasmic domain of ACE2 revealed a conserved consensus calmodulin-binding motif [59]. Moreover, a study demonstrated that calmodulin binds a 16-amino acid synthetic peptide corresponding to residues 762–777 within the cytoplasmic domain of human ACE2, forming a calcium-dependent CaM–peptide complex [62]. In this regard, a study in vitro demonstrated that CaM antagonists, such as calmidazolium, resulted in increased soluble ACE2 plasma activity. This calmidazolium-mediated stimulation of ACE2 shedding was time- and dose-dependent [59,62]. Additionally, another preclinical study demonstrated that vitamin D caused a slight increase in the level of the ACE2 plasma activity [63]. Therefore, the catalytic ectodomain shedding of ACE2 is potentially an important mechanism by which local ACE2 activity can be regulated; the shedding of ACE2 may represent a mechanism by which viral entry and infection may be controlled, such as with SARS-COV-2 [62].

A diverse array of target proteins has been identified, such as CaM kinases and ion channels, which are known to interact with CaM with or without calcium [64]. CaM also extends the reach of calcium by activating phosphorylation pathways. Calcium/CaM binding relieves the autoinhibition of the catalytic domain of CaM kinase family enzymes [58]. Some studies suggest that vitamin D plays an important role in the CaM function [65,66]. The administration of 1,25(OH)2D in vitro increased the apparent binding of CaM [65]. This increase in CaM binding coincided with the increased ability of the duodenal brush border membrane (BBM) vesicles to accumulate calcium. This mechanism may underlie the ability of 1,25(OH)2D to stimulate calcium movement across the intestinal BBM [65]. Moreover, a structural analysis demonstrated that CaM-dependent kinase IV augments 1,25-(OH)2D3/vitamin D receptor(VDR)-mediated transcription through multiple mechanisms, including by promoting the phosphorylation of VDR, increasing VDR protein expression, stimulating the intrinsic transactivation of steroid receptor coactivator-1 and increasing the interaction of VDR with steroid receptor coactivator-1 and steroid receptor coactivator-2 [66]. These multiple effects culminate in the synergistic enhancement of VDR-mediated transcription by CaM kinase IV and steroid receptor coactivator proteins [66]. In contrast, preclinical studies demonstrated the important role of ADAM-17 in sepsis as an essential regulatory event in the pro-inflammatory cytokine cascade [67]. Furthermore, a basic science study reported that endotoxins and the pro-inflammatory cytokines, interleukin-β (IL-1β) and TNFα, all enhanced soluble catalytic ectodomain ACE2 shedding from the airway epithelia, suggesting that soluble ACE2 plasma activity generation may have host defense functions [68]. Significantly, a study in vitro demonstrated that hrsACE2 can significantly block early stages of SARS-CoV-2 infections [69]. Additionally, regarding a novel hrsACE2 with a fragment crystallizable region (Fc) domain of human immunoglobulin IgG1 (ACE2-Ig) and an ACE2 variant (HH/NN), in which two active-site histidine residues (amino acids 374 and 378) had been altered to asparagine residues to reduce the catalytic activity (mACE2-1g), both have high-affinity binding to the receptor-binding domain and a neutralized virus pseudotyped with SARS-CoV-2 spike proteins in vitro [70]. Similarly, another pre-clinical study reported an improved soluble catalytic ectodomain of ACE2, termed a “microbody.” This ACE2 ectodomain is fused to Fc domain 3 of the immunoglobulin heavy chain. The protein contains an H345A mutation in the ACE2 catalytic active site that inactivates the enzyme without reducing its affinity for the SAR-CoV-2 spike protein. The ACE2 microbody protein inhibits the entry of the SARS-CoV-2 spike protein pseudotyped virus and the replication of live SARS-CoV-2 in vitro. Its potency is 10-fold higher than hrsACE2 [71]. Therefore, the administration of the exogenous soluble catalytic ectodomain of ACE2 could be a potential approach for SARS-CoV-2 therapy. Furthermore, a recombinant human ACE2, named APN01, is currently under evaluation as a treatment for COVID-19 in a phase two clinical trial (NCT04335136). In contrast, a retrospective cohort study showed that a likely deficient vitamin D status was associated with increased COVID-19 risk [72]. Moreover, many studies have shown that low vitamin levels are associated with poor prognosis in COVID-19 [73,74]. Additionally, a cohort study of the US acute care hospitals reported that statins, vitamin C or D, ACEIs, and calcium channel blockers were associated with decreased odds of death in patients with COVID-19 [75]. Moreover, a pilot randomized clinical study revealed that the administration of a high dose of calcifediol or 25-hydroxyvitamin D reduced the need for ICU treatment in patients requiring hospitalization due to proven COVID-19 [76]. Similarly, another study reported that regular bolus vitamin D supplementation was associated with less severe COVID-19 and better survival in frail elderly patients [77]. Hence, providing vitamin D for the correct physiological function of CaM to increase ACE2 shedding to generate soluble catalytic ectodomain of ACE2 by the ADAM-17 in response of the SARS-CoV-2 infection could improve the COVID-19 prognosis. Although it is uncertain whether vitamin D improves severe COVID-19 prognosis, vitamin D reposition in patients with vitamin D deficiency is a feasible and harmless adjuvant treatment for COVID-19 [76,77]. Therefore, understanding the ACE2 molecular structure and biochemical functions could provide a better understanding of the SARS-CoV-2 physiopathology and new insights for COVID-19 treatment. This notwithstanding, more studies are required to confirm the therapeutic role of the soluble catalytic ectodomain of ACE2 in COVID-19. 

## 9. ACE2 Shedding and ACE2 Cell Expression as Prognostic Markers in COVID-19

In epidemiological registries, reported smell or taste dysfunction in 56.9% of COVID-19 cases [78], and in 92.6% of mildly symptomatic cases [79]. Moreover, some studies reported that anosmia is a protective factor for hospitalization (OR = 0.42; 95% CI = 0.19 to 0.90) [80]. Additionally, anosmia is associated with lower mortality (OR: 0.180, 95% CI: 0.069–0.472) and ICU admission (OR: 0.438, 95% CI: 0.229–0.838, *p* = 0.013) [81]. In addition, anosmia was associated with lower adjusted values of D-dimer, CRP, and higher lymphocyte levels. Furthermore, anosmia had a higher frequency of cough, headache, and myalgia symptoms [81]. Other studies reported that those with headaches had a lower risk of mortality (OR: 0.39, 95% CI: 0.17-0.88, *p* = 0.007). Variables that were independently associated with higher odds of having a headache in COVID-19 patients were anosmia, myalgia, and female sex. Conversely, variables that were associated with lower odds of having headache were increased CRP, D-dimer, and lymphopenia on admission [82]. Contrastingly, a structural analysis demonstrated that neuropilin-1 (NRP1) was a cofactor for SARS-CoV-2 infection [83], and when both ACE2 and TMPRSS2 were present, NRP1 gave an additional increase [83]. This was due to an increased virus uptake into the cell rather than an increase in virus binding to the cell surface [84]. Hence, cells expressing ACE2 and TMPRSS2 are indispensable for SARS-CoV-2 infection, and, with the addition of NRPI, enhances SARS-CoV-2 entry and infection [83,84]. NRP1 is a type I transmembrane protein that is expressed in several populations of neurons, including dorsal root ganglia neurons and spinal motor neurons [85]. Additionally, NRP1 is abundantly expressed in the respiratory and olfactory epithelium, with the highest expression in endothelial and epithelial cells [83]. NRP1 is an essential modulator of embryonic angiogenesis with additional roles in vessel remodeling and arteriogenesis. NRP1 also enhances arteriogenesis in adults to alleviate pathological tissue ischemia. However, in certain circumstances, vascular NRP1 signaling can be detrimental, as it may promote cancer by enhancing tumor angiogenesis or contribute to tissue edema by increasing vascular permeability [86]. NRP-1 is expressed in various human tumors, including prostate cancer, breast cancer, melanoma, and pancreatic adenocarcinoma [87]. NRP1 expression has been shown to increase tumorigenicity, possibly by promoting vascular endothelial growth factor-mediated angiogenesis [87]. A Chinese cohort of subjects with SARS-CoV-2 reported that 1% had a history of tumors, which was higher than the occurrence of tumors in the general Chinese population (0.29%) [88]. A report from Italy indicated that 24.5% of subjects with SARS-CoV-2 had active cancer [89]. Moreover, a New York cohort showed that patients with cancer were intubated significantly more frequently (RR 95% CI 1.89 (1.37–2.61)), but the rate of death was not significantly different [90]. Additionally, there are some reports of patients with profound immunosuppression after undergoing hematopoietic stem-cell transplantation or receiving cellular therapies who may shed viable SARS-CoV-2 for at least two months using the Centers for Disease Control and Prevention guidelines [91]. Similarly, a tertiary care hospital cancer reported that SARS-CoV-2 clearance times differ substantially depending on the criteria used, and may be prolonged in cancer patients [92]. This suggests the higher expression of NRP1 in oncological patients could have implications for SARS-CoV-2 clearance. Nonetheless, more studies are needed to confirm whether oncologic patients had prolonged SARS-CoV-2 PCR and their clinical implications. On the other hand, NRP1 has been implicated in Kallman syndrome, a congenital disease characterized by hypogonadism and anosmia. Dysregulation in axonal guidance, caused by a malfunction in the interaction of NRP1 with semaphorin 3A (SEMA3A), leads to the development of this syndrome [93]. Furthermore, there are many reports of anosmia cases with abnormal enhancements on olfactory bulbs in patients with anosmia, and all of them were non-severe COVID-19 [94,95,96]. In contrast, structural analysis found that although the human brain organoid system express a low level of ACE2, the human neurons are indeed a target for SARS-CoV-2 [97]. The low ACE2 expression in olfactory cells and neuronal cells [83], but with the highest level of ACE2 in olfactory bulbs compared with that in the hippocampi and the brain cortex [63], suggests that the neurotropism of SARS-CoV, to some extent, could be that it depends not only on ACE2, but rather needs other cofactors, such as NRP1, which is abundantly expressed in the respiratory and olfactory epithelium [97]. Moreover, the low nasal congestion (3.7%) and the rare rhinorrhea in COVID-19 cohorts [78] supports the idea that there is neurological dysfunction in the olfactory bulbs, suggesting that NRP1 could be a possible candidate for the physiopathology mechanism of the anosmia in COVID-19. On the other hand, an evidence-based review of cutaneous manifestations of COVID-19 showed that acral lesions resembling pseudo-chilblains were the most frequent lesion identified (40.4% of cases), appearing in young adults (mean age, 23.2 years) after the onset of extracutaneous COVID-19 symptoms [98]. Many prospective case series suggest that chilblain-like lesions are mainly on the toes and are associated with mild or asymptomatic SARS-CoV-2 infection [99]. The real-time PCR testing results were negative in all cases, and SARS-CoV-2 serology results were positive in 30%. D-dimer concentration levels were elevated in 60%. Cryoglobulinemia and parvovirus B19 serologic results were negative for all tested patients. The course of chilblains was favorable in all cases, with complete healing of the lesions, but 35% of the patients had cold toes or acrocyanosis at a median follow-up of 27 days [100]. In contrast, type I interferons are crucial in the early response to viral infections, and a study reported a significantly higher interferon-alpha (IFN-α) response in the patients with chilblains compared with those with moderate or severe COVID-19. Given that the production of IFN-α is higher in infancy and young adulthood, and then decreases with age, and because chilblains were never reported in the literature in any of the moderate and severe forms of COVID-19 [100], this suggest that IFN-α response could be implicated in the physiopathology of the chilblain-like lesions and are a good prognostic marker of COVID-19, and probably in these patients, SARS-CoV-2 is completely suppressed before a humoral immune response is induced [99,100]. Significantly, ACE2 can also cleave des-Arg bradykinin, neurotensin 1–13, and kinetensin. Importantly, des-Arg bradykinin is involved locally in vessel dilation through binding to the B1 receptor, while kinetensin stimulates mast cell degranulation and vascular permeability [15], and he increased ACE2 shedding could cause dysregulation of the kallikrein–kinin system and have a possible influence on cutaneous COVID-19. Nevertheless, more studies are required to elucidate the physiopathology of chilblain-like lesions in COVID-19. Hence, many studies showed that cutaneous manifestations of COVID-19, such as chilblain-like lesions and anosmia, are associated with good prognosis, reporting lower D-dimer and CRP levels, as well as lower ICU admission and mortality rates in patients with cutaneous manifestations of COVID-19, such as chilblain-like lesions and anosmia. Moreover, in these patients, the majority were young and without cardiovascular risk factors [80,81,82,94,95,96,98,99,100].Given that preclinical studies had demonstrated that ACE2 cell expression is decreased with aging [63], hypertension [92], and DM [101], and these are risk factors for developing severe COVID-19 [10,52], the high ACE2 cell expression reserve, as well as the capacity to produce an appropriate ACE2 shedding process, could be implicated in the COVID-19 prognosis, since the ACE2/AT1-7/Mas axis regulates their inflammatory and thrombotic response to the SARS-CoV-2 infection [51]. Nevertheless, ACE2 expression and ACE2 plasma activity levels, their clinical implications, and prognostic significance needs to be clarified with further studies. From a practical perspective, anosmia and chilblain-like cutaneous lesions can be used for diagnostic and stratified risk assessments of patients with COVID-19.

## 10. Conclusions

The ACE2 is a type I integral membrane that exits in two forms. The first is as a transmembrane domain that can be measured as ACE2 mRNA expression. The second is a soluble form of ACE2. The catalytic ectodomain of ACE2 undergoes shedding because of ADAM17, in which calmodulin mediates the calcium signaling pathway that is involved. CaM is a ubiquitous calcium-binding protein that is known to bind other transmembrane proteins and regulate their cell surface expression. The ACE2 release results in a soluble catalytic ectodomain of ACE that can be measured as soluble ACE2 plasma activity. The shedding of the ACE2 catalytic ectodomain plays a role in cardiac remodeling and endothelial dysfunction. It is unclear whether circulating soluble ACE2 in plasma indicates the cause or effect of an adaptive or maladaptive physiological process operating in HF and cardiovascular diseases. Notwithstanding, at the present time, it is proven by many studies that high soluble ACE2 plasma activity is a predictor of MACE, cardiovascular mortality, and all-cause mortality. Moreover, many studies have demonstrated that the catalytic ectodomain of ACE2 is an entry receptor for SARS-CoV-2 infection. In addition, endotoxins and the pro-inflammatory cytokines, IL-1β and TNFα, all enhanced soluble catalytic ectodomain ACE2 shedding from the airway epithelia, suggesting that the shedding of ACE2 may represent a mechanism by which viral entry and infection may be controlled, such as with SARS-COV-1 and SARS-COV-2. ACE2 plays an important role in inflammation and the thrombotic response, and its down-regulation may aggravate COVID-19 via the renin-angiotensin system, including by promoting pathological changes in lung injury. This ACE2 down-regulation induced by the cell entry of the virus may be particularly detrimental in patients with preexisting ACE2 deficiency due, for example, to older age, DM, and hypertension. Furthermore, some studies suggest that vitamin D plays an essential role in CaM function and causes a slight increase in ACE2 shedding. Hence, studies demonstrated that either enhancing the shedding of the soluble catalytic ectodomain of ACE2 or the administration of transgenic soluble catalytic ectodomain of ACE2 forms inhibit entry and the replication of SARS-COV-2, which confirm the important role of the soluble catalytic ectodomain of ACE2 against SARS-CoV-2 infection. Observational studies showed that deficient vitamin D status was associated with increased COVID-19 risk. Similarly, those studies revealed that low vitamin levels are associated with poor prognosis in COVID-19. In addition, some clinical trials reported that the administration of vitamin D treatment was associated with prognosis improvement in COVID-19. Therefore, the high ACE2 cell expression reserve and the capacity to produce an appropriate ACE2 shedding process could be implicated in the COVID-19 prognosis. Nevertheless, more studies are required to confirm the therapeutic role of the soluble catalytic ectodomain of ACE2 in COVID-19, as well the ACE2 expression, and ACE2 plasma activity levels, and their prognostic significance needs, to be clarified with further studies. 

## Figures and Tables

**Figure 1 biomolecules-12-00076-f001:**
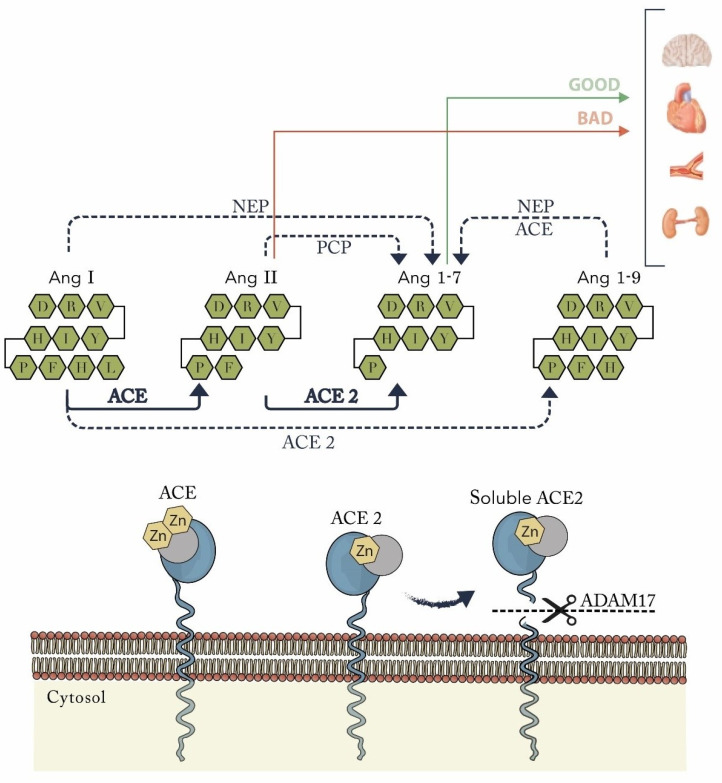
ACE2 pathway. ACE2 is >300 times more effective converting Ang II to Ang 1–7 than Ang I to Ang 1–9, in the presence of NEP or ACE, can convert Ang I and Ang 1–9 to Ang 1–7. ACE2 membrane-anchored protein at the catalytically active ectodomain undergoes shedding because of ADAM17, forming a soluble form of ACE2. ACE = angiotensin-converting enzyme; ACE2 = angiotensin-converting enzyme 2; Ang = angiotensin; NEP = neprilysin; PCP = prolylcarboxypeptidase; ADAM17 = disintegrin and metalloproteinase domain-containing protein 17; Zn = metalloproteinase zinc-binding site.

## Data Availability

Not applicable.

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
