# Peer review of "Calcium Signaling Pathway Is Involved in the Shedding of ACE2 Catalytic Ectodomain: New Insights for Clinical and Therapeutic Applications of ACE2 for COVID-19"

_biomolecules, 2022, doi:10.3390/biom12010076_

Round 1

Reviewer 1 Report

The paper is well written even if the text is not always easy to read and the paper doesn’t stay focused on its main subject ( Vit D and Covid 19).

Some revisions to manuscript are needed :

1) The title doesn’t properly reflects the subject of the text. The study is focused on ACE2 biochemical functions in Covid 19, why does the title include only the role of vit D?

2) As the main topic of the article is the role of vit D and Covid19, please integrate the TABLE 1 highlighting the effects of activated Ca2+/calmodulin-dependent kinases on vit D and ACE 2

3) Cite the studies showing an harmless effect of treatment with vitamin D (line 392-394)

Author Response

1) The title doesn’t properly reflects the subject of the text. The study is focused on ACE2 biochemical functions in Covid 19, why does the title include only the role of vit D?

We have changed the title to reflect the ACE2 biochemical functions in COVID19.

2) As the main topic of the article is the role of vit D and Covid19, please integrate the TABLE 1 highlighting the effects of activated Ca2+/calmodulin-dependent kinases on vit D and ACE 2

We required more time to modify the figure 1 to add the calmodulin pathway on ACE2, given that the computer designers that work for us are on holydays 

3) Cite the studies showing an harmless effect of treatment with vitamin D (line 392-394)

We already cite these lines, and also we review all the manuscript and added the citations in other lines too.

Reviewer 2 Report

The review, “Vitamin D regulates the calmodulin a calcium signaling path-2 way that is involved in the shedding of ACE2 catalytic ectodo-3 main: The role of vitamin D in COVID-19” discusses the close connection of ACE-2, calcium signaling, and vitamin D in COVID-19 prognosis and prospective treatment. The review shows a decent attempt to comprehensively stitch together the important information regarding role of ACE-2 in COVID-19, however, it can be improved by addressing following concerns.

  1. The title of this review indicates its focus on the role of vitamin D in COVID-19, however, vitamin D covers only 1/10th part of the manuscript and a lot of emphasis is given to ACE-2 receptor. Authors should either balance the information on ACE-2 and Vitamin D, or modify the title to justify the content of the review.
  2. Similarly, the conclusion section and the summary statement of the abstract do not provide justified emphasis on vitamin D.
  3. The references are missing supporting many statements, one example is at line 80 sentence, ”ACE2 activity is unaffected by 10 μM lisinopril, enalaprilat, or captopril, but activity was completely inhibited by 10 μM of ethylenediaminetetraacetic acid (EDTA)” require a reference.
  4. Overall, at many places, the sentences appear incomplete, example include but not limited to the first line of the abstract, “(ACE2) is a type I integral membrane receptor”.
  5. Why there is no description under the subheading “The ACE2 shedding, ACE2 cell expression, and clinical presentations associated with good prognosis in COVID-19? Either remove one subheading or merge them together to properly summarize the described text.

Author Response

  1. The title of this review indicates its focus on the role of vitamin D in COVID-19, however, vitamin D covers only 1/10th part of the manuscript and a lot of emphasis is given to ACE-2 receptor. Authors should either balance the information on ACE-2 and Vitamin D, or modify the title to justify the content of the review. We have balanced the information and we changed the title of the manuscript to adjust to the content of the review.
  2. Similarly, the conclusion section and the summary statement of the abstract do not provide justified emphasis on vitamin D. We changed the conclusion and the abstract to justified the content of the review.
  3. The references are missing supporting many statements, one example is at line 80 sentence, ”ACE2 activity is unaffected by 10 μM lisinopril, enalaprilat, or captopril, but activity was completely inhibited by 10 μM of ethylenediaminetetraacetic acid (EDTA)” require a reference. We added the cite in the line 80, also we review all the sentences and added the missing citations.
  4. Overall, at many places, the sentences appear incomplete, example include but not limited to the first line of the abstract, “(ACE2) is a type I integral membrane receptor”. We changed the sentences that appear incomplete to get a better comprehension of the information.
  5. Why there is no description under the subheading “The ACE2 shedding, ACE2 cell expression, and clinical presentations associated with good prognosis in COVID-19? Either remove one subheading or merge them together to properly summarize the described text. We resolved this mistake and removed one subheading.

Round 2

Reviewer 2 Report

The authors have satisfactorily addressed all the concerns and I do not have any further comments.